# An Interplay between Lossy Mode Resonance and Surface Plasmon Resonance and Their Sensing Applications

**DOI:** 10.3390/bios12090721

**Published:** 2022-09-04

**Authors:** Deependra Singh Gaur, Ankit Purohit, Satyendra Kumar Mishra, Akhilesh Kumar Mishra

**Affiliations:** 1Department of Physics, Indian Institute of Technology Roorkee, Roorkee 247667, India; 2Department of Electrical and Computer Engineering, Laval University, Quebec City, QC G1V 0A6, Canada

**Keywords:** optical fiber sensor, surface plasmon resonance, lossy mode resonance, indium-tin oxide, silver, sensitivity, detection accuracy

## Abstract

Conducting metal oxide (CMO) supports lossy mode resonance (LMR) at the CMO-dielectric interface, whereas surface plasmon resonance (SPR) occurs at the typical plasmonic metal-dielectric interface. The present study investigates these resonances in the bi-layer (ITO + Ag) and tri-layer (ITO + Ag + ITO) geometries in the Kretschmann configuration of excitation. It has been found that depending upon the layer thicknesses one resonance dominates the other. In particular, in the tri-layer configuration of ITO + Ag + ITO, the effect of the thickness variation of the sandwiched Ag layer is explored and a resonance, insensitive to the change in the sensing medium refractive index (RI), has been reported. Further, the two kinds of RI sensing probes and the supported resonances have been characterized and compared in terms of sensitivity, detection accuracy and figure of merit. These studies will not only be helpful in gaining a better understanding of underlying physics but may also lead to the realization of biochemical sensing devices with a wider spectral range.

## 1. Introduction

Surface plasmon resonance (SPR) is generated at metal and dielectric interface by transverse magnetic fields (TM) or p-polarized light [1]. It is impossible to excite the SPR mode by direct light due to the momentum mismatch between the SPR mode and incident light [2]. In order to efficiently excite these modes, we need a momentum matching scheme. Several such schemes have been proposed, e.g., passing light through a high RI prism, using a grating, etc. [3]. Further, there exist two coupling configurations for SPR excitation—Otto and Kretschmann [4,5]. Owing to its ease of implementation, the Kretschmann configuration is preferred often.

The growing field of SPR has attracted significant research attention over the years due to its wide range of applications, which includes nano-antennas [6], imaging [7], biosensing, and so on and so forth [8]. Several extensive theoretical and experimental studies have been conducted on SPR-based sensors in the past [9,10,11,12,13]. A sensor’s novelty is determined by the particulars of the plasmonic material used and the design implemented. Waveguide-based sensors have attracted a lot of attention due to their industrial applications. These sensors use a plasmonic material deposited as a thin film around the waveguide (e.g., an optical fiber). These materials can be classified into three categories based on the resonances they support. The first class of materials is plasmonic materials, which support SPR and have a real permittivity that is negative and larger in magnitude than both the imaginary permittivity and the permittivity of the surrounding medium. In the second category of materials, the real part of the material permittivity is positive and greater than both its imaginary part and the permittivity of the surrounding medium. The LMR phenomenon is observed in this category of materials. The third class of materials also exists for which the real part of the permittivity is close to zero and the imaginary part is large. Such a material supports long-range surface exciton-polariton [14]. The present work only focuses on the first and second classes of materials. 

LMR results out of the coupling between lossy mode and evanescent wave at a particular thickness of the thin film [15]. Only a few studies are reported on the application of LMR to sensing because the selection of the appropriate material for the thin film is critical [16,17,18,19].

Different types of waveguide structures have been utilized to realize SPR and LMR-based sensors. In particular, plasmonic fiber grating based, U-shaped and D-shaped optical fiber-based biosensor are explored extensively [19,20,21]. The simultaneous generation of LMR and SPR on the same planer platform has also been reported [22].

Indium tin oxide (ITO) is one of the CMO materials that supports LMR. It is a transparent material with an optical band gap of 3.6 eV, which restricts band-to-band transitions. The electronic and optical properties of ITO can be tuned during fabrication, resulting in a significant variation in its characteristics [23]. This property can be used to shift the resonance wavelength of LMR. Unlike SPR, the excitation of LMR has the advantage of not requiring specific polarization for incident light. Additionally, it is possible to generate multiple dips in the transmission spectrum. LMR dips are usually found in the IR and UV regions but, with the proper optimization of thin-film and with the use of other materials, they can also be observed in the visible region [14]. The IR dip is observed due to oscillations in charge density along the metal-dielectric interface. In contrast, a charge density oscillation along the thickness of the metal film is responsible for UV dip [16].

LMR is also found suitable for sensing applications. The performance and novelty of the sensor are determined by the material and sensing probe design used. Because of the excellent characteristics of optical fibers, these are being used as substrates for depositing ITO thin films to constitute the sensor [24]. 

Our present study examines the characteristics of bi- and tri-layer fiber optic sensing probes based on ITO that enable simultaneous excitations of SPR and LMR both. In bi- and tri-layer geometries, we investigate ITO + Ag and ITO + Ag + ITO structures, respectively. Applied biosensing, chemical analysis, quality assurance of food, and wavelength filtering are some of the potential applications for the proposed sensing probe.

## 2. The Model

To generate LMR, the lossy mode must be coupled with the evanescent wave. At a particular angle or wavelength, the effective index of the evanescent wave matches with the effective index of the lossy mode. The effective RI of the evanescent wave is given by
(1)$$n_{eff}=n_{p}sin\left(\theta_{i}\right)$$
where $n_{p}$ is the RI of the substrate, and $\theta_{i}$ is the incident angle of the light. This relationship shows that the RI of evanescent waves can be controlled by the incident angle of incident light and/or corresponding wavelength.

Figure 1 schematically shows the proposed sensing probe in the Kretschmann configuration. The probe consists of a multimode fiber with a core diameter of 400 μm and a numerical aperture of 0.22. The 1 cm fiber cladding has been removed from the fiber probe. On top of the unclad (exposed) core, layers of ITO and Ag have been considered. 

At one end of the fiber, light from a polychromatic source is launched, and the spectrometer records the corresponding transmission spectrum at the other end. At a certain wavelength, called the resonance wavelength, the spectrum exhibits a minimum transmitted power. A change in the sensing medium (i.e., RI of the analyte) will alter the resonance wavelength. The sensitivity of the sensor is defined as the shift in resonance wavelength corresponding to the change in the RI of the analyte. Another important characterization parameter, figure of merit (FOM) is defined as the ratio of sensitivity to the full width at half maximum (FWHM) of the transmission dip. Additionally, detection accuracy (DA) is another important parameter that measures the sharpness of the resonance. The following expression relates these parameters (sensitivity, FOM, and DA) with each other [25].
(2)$$\mathrm{FOM}=\frac{\mathrm{Sensitivity}}{\mathrm{FWHM}}=\mathrm{Sensitivity}\times \mathrm{DA}$$
where $\mathrm{DA}\propto \frac{1}{\mathrm{FWHM}}$.

Some materials are highly sensitive but have low FOM. In contrast, others have poor FOM. Hence, materials need to be carefully selected.

For multilayer structures, the transfer matrix method is used to calculate the transmission spectrum. Consider a k^th^ layer of thickness d_k_, having complex RI n_k_, and dielectric coefficient $\varepsilon_{k}$. The transfer matrix for N layer system is expressed as
(3)$$M=\overset{N}{\underset{k=2}{\prod }}M_{k}=\left[\begin{array}{cc} M_{11} & M_{12}\\ M_{21} & M_{22} \end{array}\right]=\left[\begin{array}{cc} cos\left(\beta_{k}\right) & -i\;sin\left(\beta_{k}/q_{k}\right)\\ -i\;q_{k}sin(\beta_{k}) & cos\left(\beta_{k}\right) \end{array}\right]$$
where $\beta_{k}$ and $q_{k}$ are defined as $\left(2\pi d_{k}/\lambda \right){\left(\varepsilon_{k}-n_{1}{}^{2}sin^{2}\theta_{1}\right)}^{1/2}$ and ${\left(\varepsilon_{k}-n_{1}{}^{2}sin^{2}\theta_{1}\right)}^{1/2}/\varepsilon_{k}$, respectively and $\theta_{1}$ is the incident angle of the ray, while λ is the wavelength of the incident light. The reflection coefficient $r_{p}$ of p-polarized (TM polarized) incident wave through the film is expressed as:(4)$$r_{p}=\frac{\left(M_{11}+M_{12}q_{N}\right)q_{1}-\left(M_{21}+M_{22}q_{N}\right)}{\left(M_{11}+M_{12}q_{N}\right)q_{1}+\left(M_{21}+M_{22}q_{N}\right)}$$

The reflectance, R, for TM polarized light is given as
(5)$$R={\left|r_{p}\right|}^{2}$$

A detailed description of this matrix method is given elsewhere [6,13]. The rays launched within the well-defined range of angle would be guided and the range is given by $\theta_{1}={sin}^{-1}\left(n_{cl}/n_{1}\right)$ to $\theta_{2}=\pi /2$.

The transmitted power at the output end of the fiber is given by
(6)$$P_{trans}=\frac{\int_{\theta_{1}}^{\theta_{2}}R_{p}^{N_{ref}\left(\theta \right)}n_{1}^{2}\left(sin\theta cos\theta /{\left(1-n_{1}^{2}cos^{2}\theta \right)}^{2}\right)d\theta }{\int_{\theta_{1}}^{\theta_{2}}n_{1}^{2}\left(sin\theta cos\theta /{\left(1-n_{1}^{2}cos^{2}\theta \right)}^{2}\right)d\theta }$$
where
(7)$$N_{ref}\left(\theta \right)=\frac{L}{D\;tan\theta }$$

The number of reflections occurring in the sensing region is denoted by the Equation (7), where *L* is the length of the unclad region and *D* is the diameter of the fiber. The dielectric constant of the Ag and ITO layer is calculated by the Drude dispersive model expressed as
(8)$$\epsilon \left(\lambda \right)=\epsilon_{r}+i\epsilon_{i}=1-\frac{\lambda^{2}\lambda_{c}}{\lambda_{p}^{2}\left(\lambda_{c}+i\lambda \right)}$$
and
(9)$$\epsilon \left(\lambda \right)=\epsilon_{r}+i\epsilon_{i}=3.8-\frac{\lambda^{2}\lambda_{c}}{\lambda_{p}^{2}\left(\lambda_{c}+i\lambda \right)}$$
respectively, where $\lambda_{p}$ and $\lambda_{c}$ is the wavelength corresponding to bulk plasma frequency and collision wavelength. In the case of Ag, $\lambda_{p}=0.14541\;\mu m$ and $\lambda_{c}=17.6140\;\mu m$, whereas $\lambda_{p}=0.56497\;\mu m$ and $\lambda_{c}=11.21076\;\mu m$ for ITO. The Sellmeier equation has been used to determine the RI of the fiber core [13]. We have assumed that above dispersion relations are valid in the whole wavelength range of investigation.

In order to fabricate the sensing probe, we use multimoded plastic clad fibers. The cladding can be removed by a few centimeters (a length of 1 cm of cladding is suitable for sensing applications) and then cleaned in a vacuum chamber using ion plasma bombardment and acetone. The unclad port of the fiber can be coated with metal or ITO after cleaning. Depending on the deposition techniques we have used, the uniformity of the films will vary. In order for the sensing probe to work correctly, the film uniformity must be good. A high-quality film can be achieved using sputtering and e-beam evaporation. The probe can be characterized by injecting light through one of the fiber faces and analyzing its sensing performance using a spectrometer at the other end of the fiber [26,27,28,29,30].

## 3. Results

The following two cases have been discussed in this section- in the first case, the ITO layer is deposited directly on the fiber core followed by the Ag layer (bi-layer sensing probe), and in the second case, an additional layer of the ITO is deposited over the Ag (tri-layer sensing probe).

### 3.1. Bi-Layer Configuration (ITO + Ag)

In this section, we numerically investigate a bilayer configuration of ITO + Ag coated fiber probe. In the first round of the simulation, the thickness of the ITO layer is fixed at 80 nm, whereas the thickness of the Ag layer is varied from 10 nm to 60 nm.

In the transmission spectrum, we observe two resonance dips for probe configuration with 80 nm ITO and 10 nm Ag for various values of analyte RIs, as shown in Figure 2a. Since the real part of the dielectric constant of the ITO is positive and larger than its imaginary part, at the lower wavelength region the condition of LMR generation is supported. Experimental evidence supports the LMR generation at short wavelengths and SPR excitation at long wavelengths [18]. The resonance dip in the visible region is caused by the LMR phenomena, while the second dip is the result of SPR. We would like to note here that the dielectric constant of ITO remains positive only for shorter wavelengths while at longer wavelengths it becomes negative (see Equation (9)). Therefore, for longer wavelengths, we see SPR resonance while in shorter wavelengths the probe supports LMR. These observations are well documented in the literature [14]. We would also like to note that for SPR excitation we require an interface of materials with opposite permittivity (one positive and other negative). An SPR dip can be observed even for a thin layer of Ag (i.e., 10 nm). Figure 2b illustrates that the SPR dip is more pronounced at a larger thickness of Ag. The thickness of the Ag layer, therefore, plays a critical role in the development of the SPR dip. In addition, as Ag thickness increases, the SPR dip becomes less sensitive to the RI variations in the analyte, while the LMR dip’s sensitivity increases.

The thickness of the ITO layer, however, significantly impacts the development of SPR and LMR resonances. As shown in Figure 3a, both resonances emerge with increasing thickness of the Ag layer for the 50 nm thick ITO layer. If the thickness of the ITO layer is less than 50 nm, SPR and LMR resonance dips still grow with the thickness of the Ag layer as shown in Figure 3b. Both of these cases show that SPR dip is insensitive to the RI of the sensing medium (transmitted power plot not shown).

In Figure 4a, we have plotted the LMR and SPR dip sensitivity against the thickness of the Ag layer for 80 nm thick ITO film. Both resonances are initially sensitive to changes in RI, but their sensitivities are drastically influenced by the thickness of the Ag layer. Nevertheless, the LMR sensitivity is improved as a result of the thicker Ag layer, while the SPR sensitivity is reduced as depicted in Figure 4a. At a very large thickness of the Ag layer, the SPR dip becomes insensitive to any RI variation of the analyte. This insensitive dip can be used as a reference point for characterizing the sensor’s performance. The variation in the resonance wavelength of the LMR and SPR dip is also shown in Figure 4b. As Ag thickness is increased, the SPR resonance wavelength shifts slowly towards the smaller wavelength side. Also, shown is the transmitted power at the resonance wavelengths in Figure 4c. From the figure, we observe that the wavelength that corresponds to the LMR transmitted power minimum decreases to a minimum at a particular thickness of the Ag layer, and then increases. In contrast, the transmitted power for SPR dip is shifted toward the lower wavelength side with an increasing layer thickness of Ag. Previously a similar study is reported in [13], where resonance dip, observed in the visible region, was found useful for sensing applications and the second dip that appeared in the NIR region was insensitive to the surrounding RI, but we will not focus on this insensitive SPR dip here, since it is already detailed nicely in the literature [13].

Further, we have studied the sensitivity and DA of both the resonances as a function of the analyte RI for different values of the Ag layer thickness as shown in Figure 5 and Figure 6. The sensitivities of both the modes (LMR and SPR) increase with an increasing RI of the sensing medium as shown in Figure 5a,b. These plots also suggest that LMR dip is far more sensitive as compared to SPR dip. Figure 6 shows the corresponding DA variation as a function of the thickness of the Ag layer. DA for LMR dip (Figure 6a) is also relatively large as compared to that for SPR dip (Figure 6b). Also, note the opposite trends in Figure 5 and Figure 6 with variations in the Ag layer thickness. Hence there is a trade-off between optimum values of sensitivity and DA for designing the bi-layer sensing probe.

### 3.2. Tri-Layer Configuration

In this section, we investigate the tri-layer configuration (ITO + Ag + ITO). In the following tri-layer configuration, SPR is found to be more sensitive than the LMR dip. The following sub-sections analyze two important cases.

#### 3.2.1. ITO (10 nm) + Ag (10 nm) + ITO (X nm)

This configuration examines the resonance characteristics of a tri-layer ITO + Ag + ITO coated fiber sensor, where a 10 nm layer of ITO is considered on the fiber core that is followed by a 10 nm thick layer of Ag, and then a third layer of ITO with varying thickness. Two resonance dips appear in the transmission spectra when the third layer of ITO is 10 nm thick, as shown in Figure 7a. The resonance dip in the visible range corresponds to LMR and the second dip to SPR. Increasing the thickness of the third ITO layer (40 nm) causes a new resonance dip to appear in the near-infrared region, known as LMR. The newly developed LMR dip (middle dip) is not affected by the change in RI of the surrounding medium, as shown in Figure 7b. Compared to the first LMR dip, the SPR dip (third dip) shows much better sensitivity.

The occurrence of the new LMR as a function of the thickness of the third ITO layer is shown in Figure 8a,b, where the sandwiched Ag layers are kept 10 nm and 20 nm thick, respectively. Figure 8a,b illustrates that increase in the thickness of the Ag layer from 10 nm to 20 nm shifts the resonance wavelength of the first LMR and SPR dips toward the longer wavelength side. However, the resonance wavelength of the insensitive LMR dip (central dip) remains nearly unchanged. Also, it seems that with the ITO layer thickness variation, the central dip merges with SPR dip. Alternatively, it also suggests a switch-over behavior between the two dips. This observation requires further exploration.

As the thickness of the third ITO layer increases, the LMR dip’s sensitivity decreases, whereas SPR’s sensitivity increases as shown in Figure 9. The sensitivity variations of SPR and LMR dips in the tri-layer case are opposite to those in the bi-layer case (see Figure 4a).

Additionally, the DAs for the first LMR and SPR dip are shown in Figure 10a,b, respectively, for varying thicknesses of the third ITO layer. The DA for LMR decreases with sensing medium RI while the opposite trend is observed for SPR dip in Figure 10b.

Furthermore, the variation of the FOM with sensing medium RI for two dips is depicted in Figure 11a,b. The figure clearly shows the better performance of the SPR dip.

#### 3.2.2. (ITO (50) + Ag (X) + (ITO (50))

This section presents the normalized transmission spectrum for a tri-layer configuration with an Ag layer sandwiched between two ITO layers of thickness of 50 nm each. In this configuration too, two dips of LMR and one of SPR are observed, and the characteristics of these resonances are dependent on the thickness of the Ag layer, as shown in Figure 12a,b. Figure 12a shows transmittance variation with change in the sensing medium RI, while in Figure 12b, the RI of the sensing medium is kept fixed at 1.33 RIU and the thickness of the Ag layer is varied. Figure 12a depicts that SPR dip is relatively more sensitive and central LMR dip is completely insensitive. Figure 13 indicates that on further increase in the thickness of the Ag layer, the resonance wavelength of SPR dips shifts toward the shorter wavelengths; however, the resonance wavelength of the first LMR dip shifts toward the longer wavelength. The LMR dip observed in the NIR region is insensitive to the analyte RI variations. Moreover, the increased thickness of the Ag layer tends to annihilate the SPR dip as shown in Figure 13.

Further, we study the DA, sensitivity, and FOM of each resonance dip. We demonstrate that the DA, sensitivity, and FOM of the first LMR are improved as the Ag layer thickness is increased, as illustrated in Figure 14a,b,c, respectively.

This investigation is extended to the third resonance dip, as depicted in Figure 15. With the increase in the thickness of the Ag layer, DA, sensitivity, and FOM of SPR dip are improved as shown in Figure 15a,b,c respectively.

Furthermore, we have plotted the absolute square of the electric field component along the interface across all the thicknesses of probes in two configurations- ITO (10 nm) +Ag (10 nm) +ITO (40 nm) and ITO (50 nm) +Ag (10 nm) +ITO (50 nm). The corresponding wavelength values are given in figure captions. These figures clearly show a large enhancement in the field at SPR resonance (see Figure 16c and Figure 17c). These also corroborate the observed high sensitivity for SPR resonance.

We would like to note that since in the present study a plastic-clad highly multimode fiber is considered, the proposed probe is only good for room temperature applications. Although slight variations in temperature do not influence the sensor performance, at high temperatures the fiber cladding will melt down.

## 4. Discussion and Conclusions

In conclusion, ITO + Ag-based bi-layer and tri-layer fiber-optic sensors have been studied. In the case of bi-layer geometry, two modes of resonances are possible. These are called LMR and SPR, and these resonances can be used for sensing purposes. In this bi-layer configuration, the LMR dip shows better sensitivity compared to the SPR dip. DA of the LMR dip is also far better than that of the SPR dip. As the thickness of the Ag layer increases further, the SPR dip becomes insensitive and only the LMR dip can be used for sensing. We suggest that at this thickness, the SPR dip can work as a reference, and this turns the sensor into a self-referenced sensor. By choosing the appropriate thickness of Ag, this configuration can be used in chemical, and bio-sensing, whereas the same configuration can also be utilized in wavelength filtering.

Further, two configurations of tri-layer geometry are explored, wherein one SPR and two LMR dips have been observed. In the first tri-layer configuration, the thickness of the outmost ITO layer was varied, the first LMR dip that arises in the visible region is found less sensitive as compared to the SPR dip. The other LMR dip (middle dip) that appears in the NIR region is found insensitive to any change in analyte RI. This insensitive LMR dip appears if the thickness of the third layer of ITO is increased. In the second configuration, the thickness of the Ag layer was varied. Particularly, it has been shown that the resonance wavelength of SPR dip shifts toward the shorter wavelength side; however, the resonance wavelength of the LMR dip shifts toward the longer wavelength side. Furthermore, we have plotted the electric field component along the interface across all the thicknesses of probes in two tri-layer configurations to demonstrate field enhancement. The observation of the insensitive second LMR dip and its manipulation with ITO layer thickness variation are the main contribution of this work as this suggests a switching of resonance type between LMR and SPR. Also, this work provides design rules of ITO-based bi- and tri-layer structures which support the excitations of LMR and SPR. The results of the study are summarized in Table 1. We see from the table that SPR dip exhibit a very high sensitivity of 14 μm/RIU and good DA and FOM.

## Figures and Tables

**Figure 1 biosensors-12-00721-f001:**
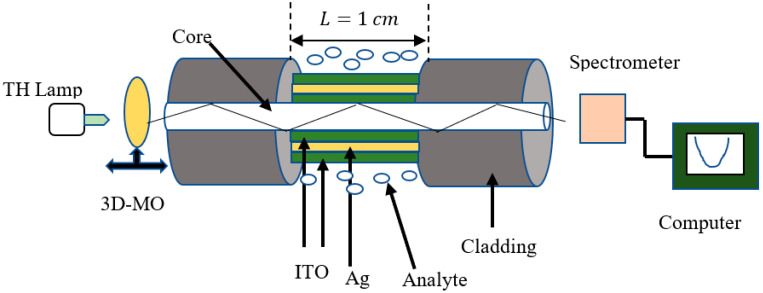
Schematic of the presented SPR setup (TH: tungsten halogen, 3D-MO: microscopic objective with 3D movement).

**Figure 2 biosensors-12-00721-f002:**
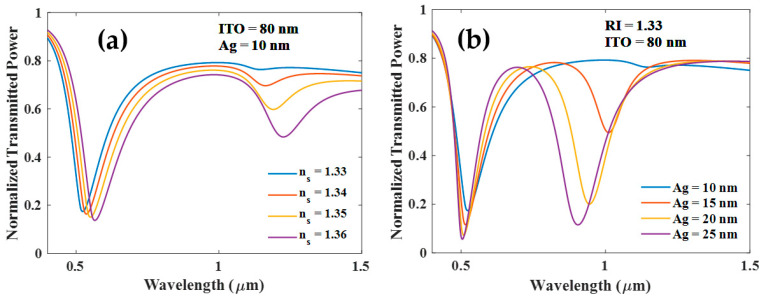
Normalized transmission spectra of sensing probe with (**a**) ITO (80 nm) + Ag (10 nm) and (**b**) for various thicknesses of Ag for 80 nm ITO and analyte RI $n_{s}=1.33$.

**Figure 3 biosensors-12-00721-f003:**
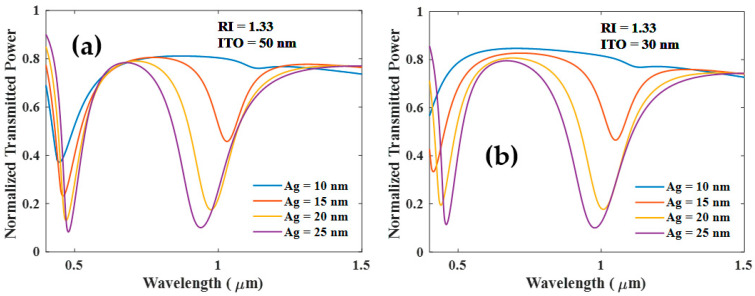
Normalized transmission spectra of sensing probe for various thicknesses of the Ag and (**a**) ITO (50 nm) (**b**) ITO (30 nm) and RI of the surrounding medium is 1.33.

**Figure 4 biosensors-12-00721-f004:**
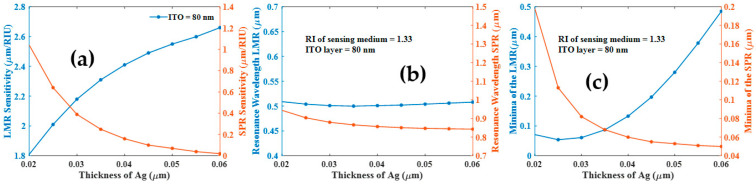
(**a**) Sensitivity (**b**) resonance wavelength and (**c**) minima of the transmission spectra at resonance wavelength of the LMR and SPR modes with the variation of Ag layer when ITO (80 nm) and RIs of the analyte are 1.33 and 1.34.

**Figure 5 biosensors-12-00721-f005:**
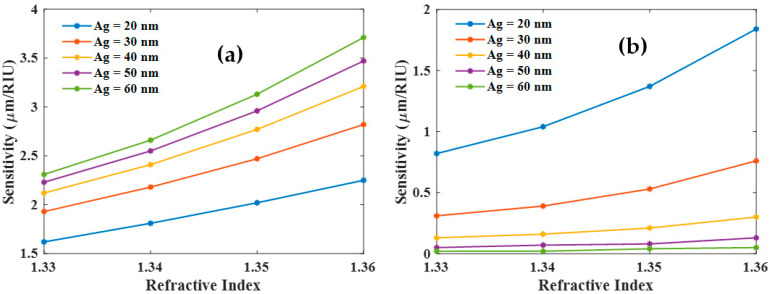
Sensitivity of the (**a**) LMR and (**b**) SPR dip with the variation of RI of the sensing medium for different thicknesses of Ag layer and 80 nm thickness of ITO layer.

**Figure 6 biosensors-12-00721-f006:**
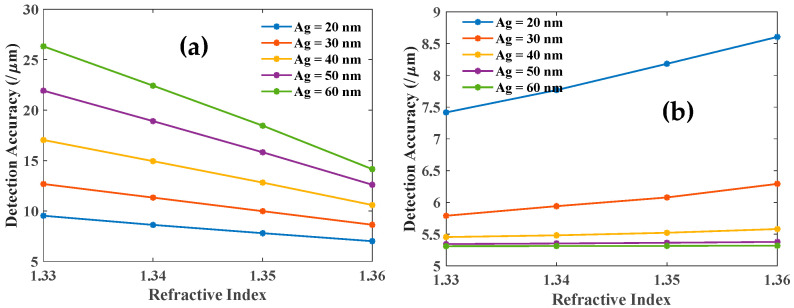
Detection accuracy of the (**a**) LMR and (**b**) SPR dip with the variation of RI of the sensing medium for different thicknesses of the Ag and 80 nm thickness of the ITO layer.

**Figure 7 biosensors-12-00721-f007:**
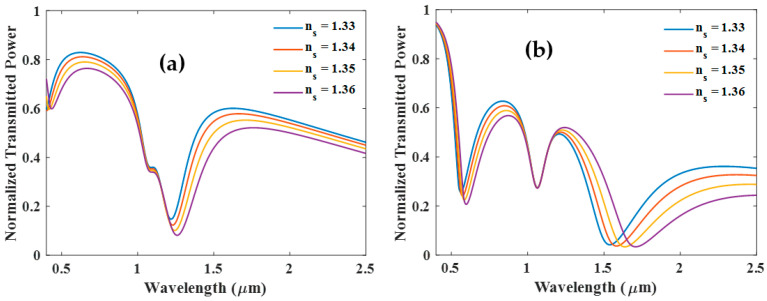
Normalized transmission spectra of sensing probe with (**a**) ITO (10 nm) + Ag (10 nm) + ITO (10 nm) and (**b**) ITO (10 nm) + Ag (10 nm) + ITO (40 nm).

**Figure 8 biosensors-12-00721-f008:**
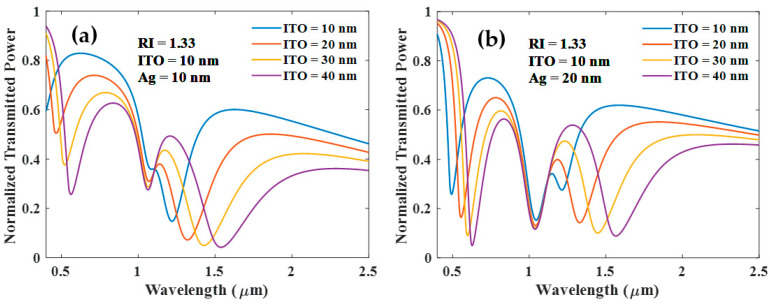
Normalized transmission spectra of sensing probe with (**a**) ITO (10 nm) + Ag (10 nm) + ITO (X nm) and (**b**) ITO (10 nm) + Ag (20 nm) + ITO (X nm).

**Figure 9 biosensors-12-00721-f009:**
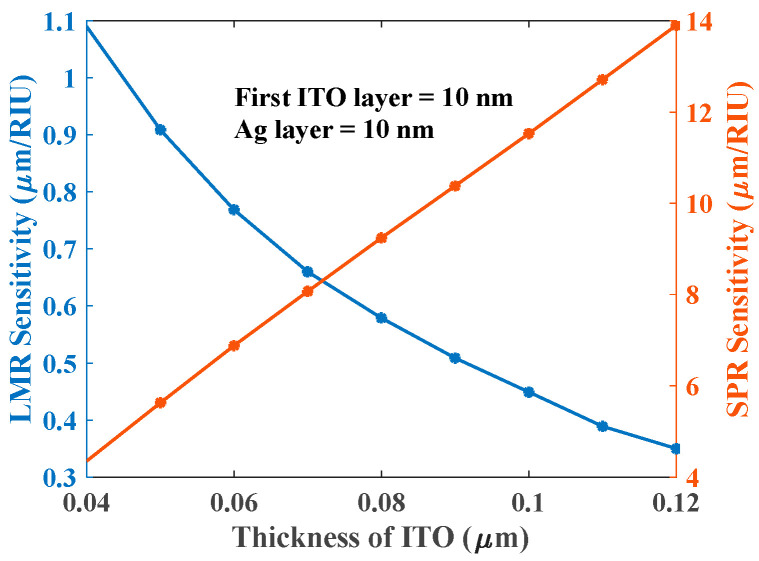
Sensitivity of the First LMR and SPR dip with the variation of the third ITO layer when the first layer is ITO (10 nm) and the second layer is Ag (10 nm) and RI of the sensing medium varies from 1.33 to 1.36 RIU.

**Figure 10 biosensors-12-00721-f010:**
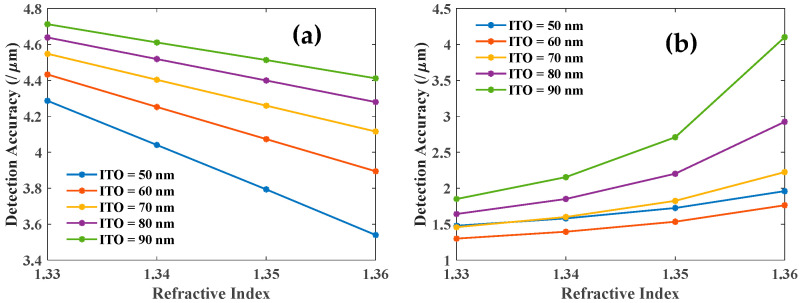
Detection accuracy of the (**a**) first LMR and (**b**) SPR dip with the variation of third ITO layer for 10 nm thick first layer ITO and 10 nm thick Ag layer.

**Figure 11 biosensors-12-00721-f011:**
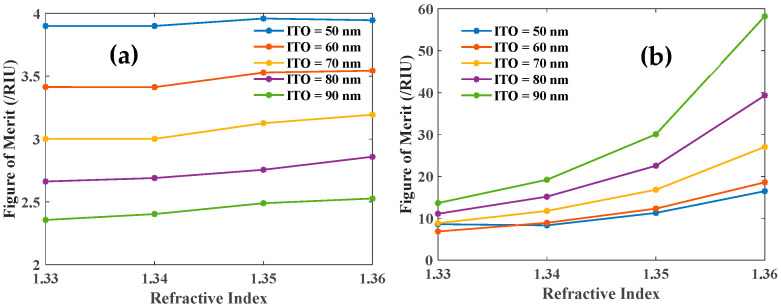
FOM of the (**a**) first LMR and (**b**) SPR dip with the variation of the third ITO layer thickness where the first layer is ITO (10 nm) and the second layer is Ag (10 nm).

**Figure 12 biosensors-12-00721-f012:**
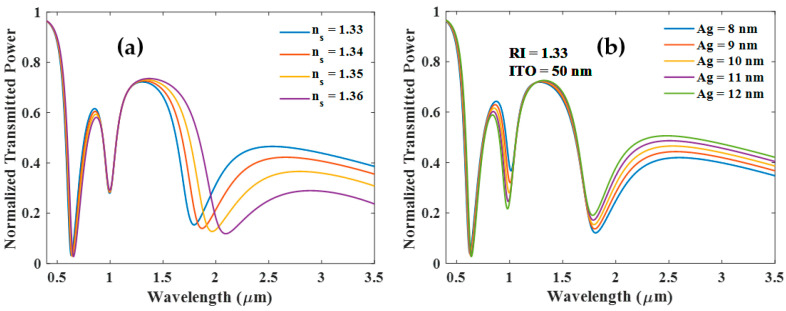
Normalized transmission spectra of sensing probe with (**a**) ITO (50 nm) + Ag (10 nm) + ITO (50 nm) and (**b**) for varying thickness of the Ag layer when both ITO layers are 50 nm thick.

**Figure 13 biosensors-12-00721-f013:**
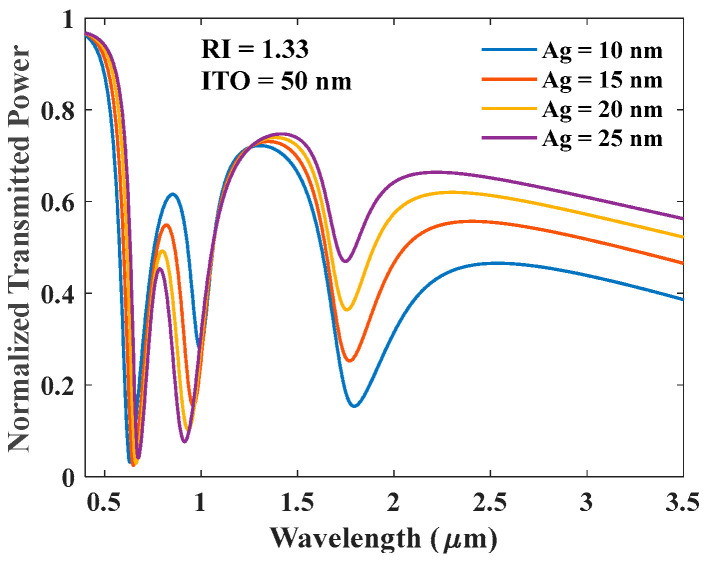
Normalized transmission spectra of sensing for varying thickness of the Ag layer when both ITO layers are 50 nm thick.

**Figure 14 biosensors-12-00721-f014:**
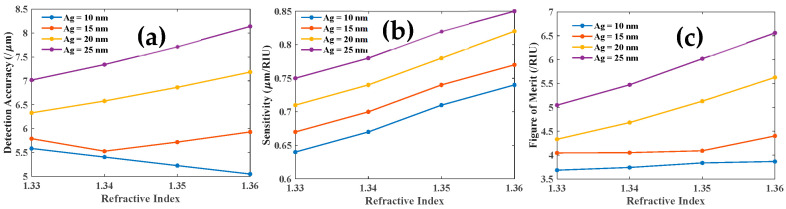
(**a**) DA, (**b**) sensitivity, and (**c**) FOM of the first LMR dip as a function of RI of the analyte.

**Figure 15 biosensors-12-00721-f015:**
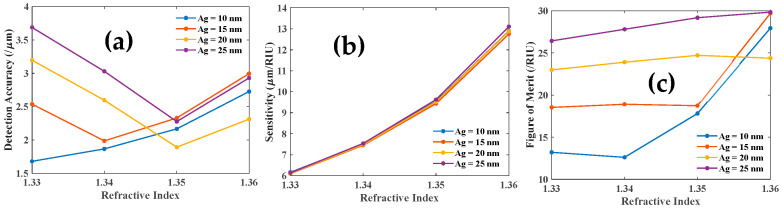
(**a**) DA, (**b**) sensitivity and (**c**) FOM of the SPR dip as a function of RI of the analyte.

**Figure 16 biosensors-12-00721-f016:**
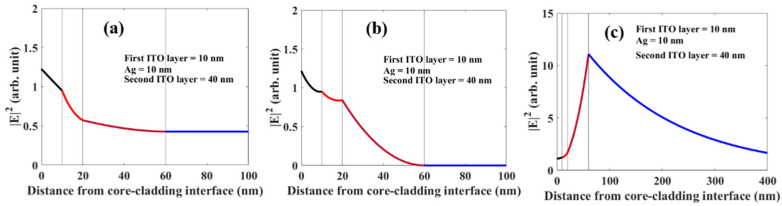
Electric field distribution corresponding to (**a**) first LMR (**b**) second LMR (**c**) SPR at their respective resonance wavelength at 561, 1065, and 1532 nm respectively for probe configuration ITO (10 nm) +Ag (10 nm) +ITO (40 nm).

**Figure 17 biosensors-12-00721-f017:**
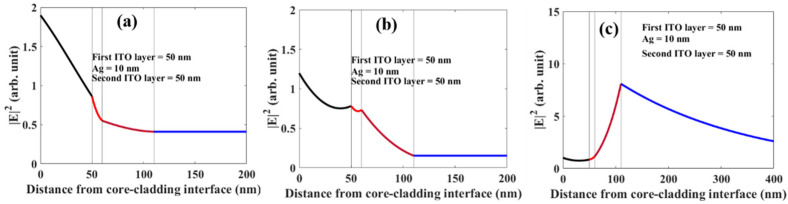
Electric field distribution corresponding to (**a**) first LMR (**b**) second LMR (**c**) SPR at their respective resonance wavelength at 631, 995, and 1798 nm respectively for probe configuration ITO (50 nm) +Ag (10 nm) +ITO (50 nm).

**Table 1 biosensors-12-00721-t001:** Summary of the results obtained in bi layer and tri layer configurations.

Configuration	Wavelength of Operation μm	Refractive Index Range	Sensitivity (μm/RIU)	DA $\left({\mu m}^{-1}\right)$	Figure of Merit $\left({\mathrm{RIU}}^{-1}\right)$
ITO(10)/Ag(X)	0.4–0.8 (LMR)	1.33–1.36	~2.6 (X=60 nm)	~26 (X=60 nm)	
	0.8–1.5 (SPR)	1.33–1.36	~1 (X=20 nm)	~7.5(X=20 nm)	
ITO(10)/Ag(10)/ ITO(X)	0.4–0.8 (LMR)	1.33–1.36	~1.1 (X=40 nm)	~4.7(X=90 nm)	~4 (X=50 nm)
	1.2–2.5 (SPR)	1.33–1.36	~14 (X=120 nm)	~4(X=90 nm)	~60 (X=90 nm)
ITO(50)/Ag(X)/ ITO(50)	0.4–0.7 (LMR)	1.33–1.36	~0.7 (X=25 nm)	~7(X=25 nm)	~5 (X=25 nm)
	1.2–3 (SPR)	1.33–1.36	~14 (X=25 nm)	~3.5(X=25 nm)	~26 (X=25 nm)

## Data Availability

The data that support the findings of this study are available from the corresponding author upon request.

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
