# Peer review of "An Interplay between Lossy Mode Resonance and Surface Plasmon Resonance and Their Sensing Applications"

_biosensors, 2022, doi:10.3390/bios12090721_

Round 1

Reviewer 1 Report

The authors have presented a novel theme of interplay between LMR and SPR for biosensing applications. This reviewer finds this paper very interesting and important for the state-of-the-art in the related area. there are some minor comments:

1. The authors can discuss the effect of possible roughness of concerned layers (ITO Ag etc.) on sensor's performance. It may not require any new simulations.

2. The authors can also incorporate some information and references about the possible experimental steps/limitations vis-a-vis the proposed LMR-SPR based sensor design.

3. What can be the possible influence of hybrid modes and/or skew rays in the fiber on the proposed design?

4. The authors may compare the sensor's performance with the existing relevant sensors based on similar themes such as LMR, SPR etc.

Author Response

Attached the file

Reviewer 2 Report

In this manuscript, Deependra Singh Gaur et. al, report the characteristics of bilayer and trilayer fiber optic sensing probes based on ITO that enable SPR and LMR to be excited simultaneously. By choosing the appropriate thickness of Ag, the thickness is also found to ensure that a particular resonance dominates the other. The overall author has to explain all the technical details, in brief, the different architectures of experiments above mentioned are convincible, and the journal on sensing engineering is also acceptable and useful as the demonstration of the sensing community. Despite listing systematically the application of LMR in sensors, need improvement in the text and the graphs, and the overall presentation of the paper. I recommend the publication of this manuscript after some minor concerns raised below:

1.        It is suggested the authors measure and report the impact of temperature variation on the sensor response. This could evaluate the practical application of the proposed sensor.

2.        I recommend that the authors give an additional explanation of phenomena in the SPR and LMR resonance wavelength has obvious shift with the combination of tri-layer structure of ITO+Ag+ITO, how in a change of refractive index on the surface of optical biber.

3.        Table 2. without any data included, i recommend the authors should check again and modify it in the revised manuscript.

4.      In addition to the biological techniques for optical assay mentioned in the introduction part, other optical detection biosensors should also be summarized, such as U-shaped fiber biosensor for MicroRNA;  real-time immunosensor of microchannel long-period fiber grating; I suggest adding some latest related reports published in the literature.

5.      In the varied thickness of the Ag layer. It has been shown that the resonance wavelength of SPR shifted toward a longer wavelength, and the resonance wavelength of the LMR dip shifted toward a shorter wavelength. please explain the phenomenon caused by what kind of reason and add it in the revised manuscript.

6.      Several typos or misspellings were found in the manuscript.

- In Figure 1, the “Analyte” should be corrected or modified.

- In line123 equation, the ? and ?1 should be defined.

- In Figure. 3 Normalized transmission spectra of sensing probe with (a) ITO (80 nm)+Ag (50 nm) and (b) ITO (80 nm) and various thickness of Ag. But the figure present ITO (50 nm) in Fig.3(a) and ITO (30 nm) in Fig 3.(b) that should be corrected or modified.

7.    The authors state “At a small thickness of the Ag layer, the LMR dip shows better sensitivity compared to the SPR dip. DA of the LMR dip is also far better than the SPR dip.” But the manuscript does not clearly state the role of SPR dip in the sensor. The authors should explain and add it in the revised manuscript.

8.   "As the thickness of the Ag layer increases further, the SPR dip becomes insensitive and only LMR dip can be used for sensing.” That mentions the Ag layer thickness affects SPR dip even disappear, suggesting the author can provide how allowed for the transmission and reflected light to be estimated sensors utilize LMR dip. This is a very important theory for sensors.

Author Response

Attached the file

Reviewer 3 Report

The work by D. S. Gaur et al deals with a numerical analysis concerning bi-layer and thee-layer structures supporting the excitation of plasmonic modes and lossy mode resonances in the Kretschmann configuration.

My main concern is about the novelty and the impact of the results, as better clarified below.

The ‘novel’ aspect, i.e. the analysis about the interplay between SPR and LMR, seems to not bringing out relevant results, or at least it is not clearly discussed. First, it is not clear how the authors demonstrate the effective LMR nature of the central dip arising in the three layer configuration. I would be curious to see the field distribution associated to this resonant effect. In addition, its insensitivity to external refractive index is that mostly worries me. In this regards, the authors conclude the work stating “The observation of the insensitive second LMR dip is the main contribution of this work”, however, neither from my personal experience, nor from the authors discussion in the present manuscript, I really do not find any advantage of exciting an insensitive dip. Typically, the effort is devoted to outperform the sensitivity with respect to the state of art…

Moreover, in my humble opinion, most of the manuscript deals with themes that are well-known within the community working in this field. This aspect is particularly evident in the introduction, which is rather didactic, reporting on assested concepts regarding SPR and LMR. The rest of the paper is basically a parametric analysis, aimed at showing the spectral characteristics as a function of geometrical parameters (basically thicknesses). The treatment is based on observations that, in some cases are by now quite obvious, in other cases they are not sufficiently supported by the data. For example, between lines 161-162, I read “The resonance dip occurs in the visible region caused by the LMR phenomena, while the second dip is the result of SPR.” How is it demonstrated? Fields maps, for example evaluated by FEM or FTDT methods would be helpful in this sense.

Concerning the aim and scope, although the title refers to biosensing applications, the paper is mostly focused on the physical aspects behind the excitation of resonant modes. Although in principle these electromagnetic effects could be used for biosensing applications, the carried out analysis, from a sensing point of view, is based on the wavelength shift as a function of what the authors define “sensing medium” external refractive index. What exactly is this “sensing medium”? If it refers to the external refractive index, this is quite far from biosensing. Typically a thin layer (few nm) attached on the sensitive area is considered to mimic the molecular binding. 

Finally, some statements are quite confusing, or at least not supported by appropriate references. For example:

Line 51: The authors state “It has been found that indium tin oxide (ITO) is the only CMO material to excite LMR.”. What about AZO, IGZO, SNO2, for example?

Lines 57-61: These statements are not really clear. LMR wavelength can change continuously, so what is the necessity of making this distinction among UV, Visible, IR regions? Moreover, oscillations at metal-dielectric interface typically refers to plasmonics…

To conclude, the aim and scope of this work is not really clear. The results discussed rather obvious, and in some points the treatment is confusing. The interplay between LMR and SPR, which should be the main topic of this work (at least according to the title) is not fully and clearly analyzed, and seems to not introduce any advantage with respect to the state of art. For this reason I do not feel confident in suggesting this work for a publication in Biosensors. 

Author Response

Attach the file

Round 2

Reviewer 3 Report

I appreciate the authors' effort in improving the manuscript clarity on the basis of the reviewer comments however some aspects still appear confusing. For example, line 231 seems to be -maybe apparently- in contrast with line 342.

Line 231: In the following tri-layer configuration, SPR is found to be more sensitive than the LMR dip.

Line 342: In the first tri-layer configuration, the thickness of ITO layer was varied, the first LMR dip that arises in the visible region is found more sensitive as compared to the SPR dip.

I invite the authors to check these points, in the indicated lines and through the whole text, better clarifying the key argumentations, in order to improve the readability.

Other aspects (cfr for example the statement in line 177 of the revised version) still miss a detailed discussion in the manuscript, although this discussion has been provided in the reply.

Moreover, I still believe that the concept of 'interplay' between two different modes (LMR and SPR), which appears in the title, is not fully discussed, as also pointed out in the first revision step. In this regard, I invite the authors to better clarify the aim and scope of the work, and to better define the impact of the results. The authors still state in the conclusions that 'The observation of the insensitive second LMR dip is the main contribution of this work' (cfr line 351). In contrast, in my humble opinion, this could be seen as an interesting work that provides design rules of bi and three layer structures based on ITO supporting the excitation of LMR and SPR. Therefore, my suggestion, as a minor revision, is to try to resume and highlight these design rules in the conclusion section.

Author Response

The attachment can be found here.
